# Irrational Beliefs about COVID-19: A Scoping Review

**DOI:** 10.3390/ijerph18199839

**Published:** 2021-09-22

**Authors:** Federica Maria Magarini, Margherita Pinelli, Arianna Sinisi, Silvia Ferrari, Giovanna Laura De Fazio, Gian Maria Galeazzi

**Affiliations:** 1Department of Biomedical, Metabolic and Neural Sciences, School of Specialization in Psychiatry, University of Modena and Reggio Emilia, 41124 Modena, Italy; federica.maria@hotmail.com (F.M.M.); margheritapinelli90@gmail.com (M.P.); arianna.sinisi@gmail.com (A.S.); silvia.ferrari@unimore.it (S.F.); 2Section of Clinical Neurosciences, Department of Biomedical, Metabolic and Neural Sciences, University of Modena and Reggio Emilia, Largo del Pozzo 7, 41124 Modena, Italy; 3Department of Law, University of Modena and Reggio Emilia, 41121 Modena, Italy; defagi31@unimore.it; 4Dipartimento di Salute Mentale e Dipendenze Patologiche, USL-IRCCS di Reggio Emilia, 42122 Reggio Emilia, Italy

**Keywords:** COVID-19, coronavirus, infodemic, misinformation, conspiracy theories, public health, mental health, social and public trust

## Abstract

Since the emergence of the recent Coronavirus Disease of 2019 (COVID-19) and its spread as a pandemic, there has been a parallel spread of false and misleading information, known as an infodemic. The COVID-19 infodemic has induced distrust in scientific communities, governments, institutions and the population, and a confidence crisis that has led to harmful health behaviours, also impacting on mental health. The aim of this study is to provide a scoping review of the scientific literature about COVID-19-related misinformation and conspiracy theories, focusing on the construction of a conceptual framework which is useful for the interpretation of the conspiracy theory phenomenon surrounding COVID-19, and its consequences. Particular socio-environmental conditions (i.e., low educational level, younger age), psychological processes and attitudes (such as low levels of epistemic trust, the avoidance of uncertainty, extraversion, collective narcissism, and a conspiracy-prone mindset), and contextual factors (e.g., high levels of self-perceived risk and anxiety) seem to underpin the adherence to beliefs that are not solely the domain of paranoids and extremists but a widespread phenomenon that has caused important health, social and political consequences.

## 1. Introduction

The pandemic spread of COVID-19 has been associated, since late 2019, with a crisis of health systems and economies in parallel with the similarly globally spreading of often conflicting information regarding the COVID-19 illness, defined by the World Health Organization (WHO) as an “infodemic” [1].

An infodemic is defined as the spread of too much information—including false or misleading information in the digital and physical environments—during a disease outbreak [1]. It causes confusion and can worsen risk-taking behaviours, with potential harmful results. It also leads to mistrust in health authorities and undermines the public health response.

Nowadays, such infodemics are aided by the ease with which misinformation—misleading information created or disseminated without manipulative or malicious intent—can be accessed and shared, especially online.

This is also exacerbated by another phenomenon: disinformation, which is the deliberate attempt of individuals, groups and organizations to confuse or manipulate, for economic or political reasons [2], or simply to disrupt public communication processes [3]. 

Focusing in detail on some of the key components of the infodemic related to COVID-19, this article aims to describe and comment on the dynamics and consequences of rumors and conspiracies about the pandemic.

The notion of what is epistemically rational or irrational has been widely discussed in the philosophical literature. In the third edition of his masterly *Theory of Knowledge* (TOK), Roderick Chisholm lays out a conception of epistemology according to which our aim is to improve ourselves epistemically, to discard epistemically unjustified or irrational beliefs while acquiring ones that are justified [4]. The epistemic rationality concerns the relationship between our belief and the evidence available to us that speaks for or against the content of that belief [5]. According to Bortolotti, whereas a belief is either true or false, there are degrees of ill-groundedness and imperviousness that apply to epistemically irrational beliefs. Saying that, we will consider epistemically irrational beliefs that are either badly supported by the evidence that is relevant to their content at the time of their adoption (ill-grounded) or scarcely responsive to contrary evidence after their adoption (impervious) [5]. 

The aim of conspiracy theories (CTs) is to explain various social phenomena as the result of plots, deliberately developed by certain powerful groups, that are exceptionally effective and no less exceptionally malicious [6]. Even though CTs are related to the oversimplification and distortion of information, they quickly provide an explanation for confusing events, assigning the causes of significant social and political events to secret strategies by powerful actors [6,7]. Particular environmental conditions, social motivations, and psychological processes and predispositions have been proposed to underpin the adherence to beliefs that are not solely the domain of paranoids and extremists, but a widespread phenomenon that cuts across demographic and political differences [8], with important social and political consequences. 

This phenomenon affects a significant part of the population, reflecting the need to understand the world, to feel safe, to belong, and to feel good about oneself and one’s social groups [9]; it is important to consider it in its complexity, rather than to easily dismiss it as delusional. In fact, conspiracy beliefs have consequences: they impact behaviour in important domains of life such as health, interpersonal relationships and safety [10].

The adoption of CTs during the COVID-19 pandemic determined a strong reduction in the adherence to health prevention guidelines and governments’ provisions, causing severe negative public health consequences [11]. CTs can negatively affect human behaviour because of the sense of uncertainty and insecurity that they evoke; Marinthe and colleagues recently found that conspiracy thinking was associated with engagement with non-normative social distancing behaviours and lower compliance with confinement [12]. Frequently, CTs are related to each other, but behaviours do not always reflect CTs’ contents. For instance, health prevention behaviours, vaccination intention and pseudoscientific practices were all predicted by irrational thinking, and the most important predictor of these attitudes was believing in CTs [13]. 

Beyond the COVID-19 pandemic, several theories tried to explain people’s social-behaviours [14]: one of the most important is the “planned behaviour” theory [15,16], according to which intention is the core determinant of action. The Health Action Process Approach states that there is also a post-intentional volition phase in which people can recur to auto-regulatory strategies that modulate their behaviours [17]. Other theories are: the “prospect theory” [18], according to which people are more motivated to avoid a loss rather than achieving a profit; the “terror management theory” [19], which states that, when facing terrifying messages, people try to maintain confidence in self-esteem and secure attachment; and the “social influence theory” [20], according to which people’s behaviours, thoughts and actions are influenced by compliance, internalization and identification.

This article aims to provide a scoping review of the scientific literature on the area of COVID-19-related misinformation and CTs, focusing on the construction of a conceptual framework which is useful for the interpretation of the conspiracy phenomenon around the COVID-19 pandemic, and its consequences. A full account of every single COVID-19 CT and the collection of data on the prevalence of CTs that are currently circulating was beyond the scope of this manuscript. 

## 2. Materials and Methods

A scoping review of the literature was undertaken, according to the framework outlined by Tong and colleagues [21] posing the question “What is known from existing research about factors potentially associated with the development and the spread of irrational beliefs concerning COVID-19? What is the impact of such irrational beliefs on social behaviour and compliance with public health recommendations? What interventions could be useful to address these effects?”. The review included the following key phases: (1) identifying the research question, (2) identifying relevant papers, (3) study selection, (4) charting the data, and (5) summarizing and reporting the results.

### 2.1. Search Strategy and Data Sources 

A combination of search terms was constructed by the research team. For each of the search terms chosen, possible alternatives were considered. The PubMed, Scopus, PsycINFO, Web of Science and CINAHL databases were searched on 12 January 2021 for papers published in English with the following keywords: (minimization OR negationism OR “irrational thinking” OR “irrational belief *” OR pseudoscience OR delusion * OR “epistemically suspect” OR “overvalued idea *” OR “conspiracy theory” OR “conspiracy theories”) AND (COVID-19 OR Sars-COV-2 OR pandemic OR coronavirus).

The papers were identified by searching for titles, abstracts and keywords, and by mapping terms to subject headings. Additional relevant research articles were identified through the reference lists from the relevant papers screened. 

### 2.2. Inclusion Criteria 

Qualitative and quantitative empirical research and opinion papers were included in the review. The search was restricted to articles published in English. There were no restrictions on the publication phase status or publication date. We also included studies focused on COVID-19 CTs’ mediatic spread, as papers systematically addressing factors influencing CTs’ endorsement and related behaviours were scarce. 

### 2.3. Exclusion Criteria 

We excluded papers if they focused on CTs not related to COVID-19 (off-topic). Finally, reviews, book chapters and editorials, and papers focused on the literary and poetic analysis of the theme were also excluded. 

### 2.4. Study Selection 

The papers were retrieved and included according to PRISMA statement recommendations [22]. The retrieved articles from the original search were screened independently by three review authors (F.M.M., M.P. and A.S.) for inclusion, on the basis of their title and abstract. Duplicates were removed, and disagreements were resolved by consensus with a fourth reviewer (G.M.G.). For each selected paper, three authors (F.M.M., M.P. and A.S.) screened the full text, and extracted and summarized the data. 

### 2.5. Charting the Data 

Data extraction was performed for the following study characteristics: year of publication, first author, journal, study design, and outcomes of interest. In a thematic summary deductive approach, we identified all of the instances in which the COVID-19 CT phenomenon was analysed, and we integrated them into a conceptual framework. 

### 2.6. Collating, Summarizing and Reporting the Results 

The included studies were assessed for their common characteristics and were regrouped into thematic groups. Each study was allocated into one thematic group, or several thematic groups when it dealt with more than one topic. The findings from the studies in each thematic group were then analysed and summarised separately using an interpretive narrative synthesis method.

The synthesis of the findings was performed according to a thematic summaries method [23], breaking down the review question into “concepts”: (1) misinformation as a substrate and adjuvant in the development of COVID-19-related CTs; (2) the dispositional and situational explanations of COVID-19 CTs; and (3) the behavioural consequences of the spread of COVID-19 CTs. A quantitative meta-analysis was not conducted due to the diversity of the populations, study designs and measured outcomes. 

## 3. Results

A total of 3940 potentially relevant records were obtained from the electronic search. After duplicates were discarded, 3485 records remained. Following a review of the titles and abstracts, 3337 more records were excluded, and a further 72 were excluded after a review of the full texts, as shown in Figure 1. 

An additional 14 relevant research articles were identified through the reference lists from the included papers, and they were also included. In total, 90 papers were included in the review. Figure 1 shows the flow-chart of the article selection. What follows is a narrative summary of the findings derived from the included studies, reported under the three main headings: (1) misinformation as a substrate and adjuvant in the development of COVID-19 related CTs; (2) the dispositional and situational explanations of COVID-19 CTs; and (3) the behavioural consequences of the spread of COVID-19 CTs.

### 3.1. Misinformation as a Substrate and Adjuvant in the Development of COVID-19-Related CTs 

#### 3.1.1. The Role of Social and Mass Media

The ubiquitous spread of COVID-19 resulted with a spate of information on social media. Most platforms were used to deliver relevant news, guidelines and precautions to people. The dark side of social media was exhibited in a tsunami of fake and unreliable news that ranged from the selling of fake cures to the use of social media as a platform to launch cyberattacks on critical information systems [24,25,26].

By identifying and analysing a sample of fake news stories published in the English language between 1 January 2020 and 30 April 2020, Naeem et al., 2020, identified the main sources which disseminated the stories, and the main types of stories, confirming the central role of social media [27]: social media accounted for the spread of half of the stories about COVID-19. The other 50% of sources included multiple sources: individuals, Donald Trump, and newspapers/websites/tabloids. The three common types of misinformation relating to COVID-19 were false claims, CTs and pseudoscientific health therapies.

The characteristics of the information sources, such as the top domains and URLs, and the implied messages in the infodemic Twitter network support the claim that Twitter was a channel for the infodemic [28], as well as Instagram [29].

Even though COVID-19 is a recent event, misinformation related to it has created a set of polarized communities, with high echo-chamberness. Memon and Carley, 2020, summarized the online COVID-19 communities in the two following groups: misinformed users, or users who are actively posting misinformation; and informed users, or users who are actively spreading true information or calling out misinformation. Their analyses show that COVID-19-misinformed communities are denser, and more organized than informed communities, with a possibility of a high volume of the misinformation being part of disinformation campaigns [30].

Quantitative measurements of the critical impact of the social network infodemic during the COVID-19 pandemic show that tweets that might mislead users by redirecting them to out-of-scope and/or malicious content were initiated by users with non-reliable medical and/or relevant specialty profiles, and consequently might be disseminating misleading non-credible medical information [31,32,33].

However, social media is not solely to blame: López Cantos and Millán Yeste, 2018, showed that public radio (i.e., in Spain) contributed to the diffusion of contents without any scientific validity, and that it puts itself at the disposal of the pseudoscientific discourse [34].

As a result of the above, an ongoing challenge during the pandemic has been to reach audiences in a crowded online environment, establishing authority as a trusted source and countering misinformation [35].

#### 3.1.2. The Epistemic Mistrust: Is It Science to Blame? 

Much of the confusion at the start of the pandemic related to fundamental scientific uncertainties about the outbreak. Key information about the virus—its transmissibility and its case-fatality rate—could be estimated only with large error margins. Many expert scientists were honest about this, but, as a consequence, it created an “uncertainty vacuum” that allowed superficially reputable sources to jump in without real expertise [36].

The scientific community lacked general authoritative figures who managed to actively contrast such misinformation [37].

Studies have been rushed to publication even in well-regarded journals. Unvetted articles on pre-press servers have received enormous attention [38]. Predatory journals have been allowing anyone with the ability to pay to publish pseudoscience, which has been amplified in mainstream news sources. Marketers have been exploiting the public’s desperation for protection against COVID-19, adding a scientific sheen to dubious products [39]. Furthermore, perhaps well-meaning experts in data science have been producing a raft of arguably meaningless research, creating a distraction at best, and wasting valuable resources at worst [40].

We have faced a very important problem in terms of the reliability of science and scientific data, and the respect of ethical rules [41]. Various studies conducted without abiding by scientific research ethics committees had to be withdrawn [42].

In a globalized world, it is so easy to have such a quantity of information at one’s disposal that the risk of unclear and confused information is massive [43]. 

The medical and social crisis created by the onset of the COVID-19 pandemic has contributed to highlight the role of researchers and bioethicists, emphasizing their social responsibility in properly informing not only the academic community, but also the general public [44].

#### 3.1.3. Epistemic Mistrust’s Implications: Vaccine Hesitancy, Stigma and Risk-Taking Behaviours 

According to the WHO, although the effectiveness of vaccinations has been proven over the years, and though adverse effects to the currently available vaccinations are extremely rare, many people continue to defer immunizations for themselves and their families. This phenomenon is known as “vaccine hesitancy”, and it is a major public health problem globally [45]. 

It has been found that inconsistent communication from public health experts and officials regarding health risks directly leads to an increase in vaccine hesitancy [46]. Studies have shown that a large majority of misinformed users may hold anti-vaccine views [30,47]. 

Misinformation and CTs of COVID-19 may also facilitate COVID-19 stigma, given that they can contribute to the marking, group labelling, responsibility, and perceived peril of the disease. For instance, stigmatizing message content was also more likely to appear in tweets that contained misinformation and CTs [48].

Unfortunately, there are some well-known historical precedents about the link between misinformation, stigma and risk-taking behaviours. 

Similarly to what has happened during the COVID-19 pandemic, misinformation was widespread during the early years of the HIV epidemic, with the effects still being visible in regions to this day [49]. Many people continue to argue that HIV does not exist, or cause AIDS, and that its therapies are toxic to human health with no advantages [50]. The influence of these false arguments can be so infectious that it can influence governmental policy. This was particularly highlighted by the Mbeki South African government’s denialism of HIV in the early 2000s, and their infamous rejection of the evidence surrounding the efficacy of HIV medication. In turn, thousands of mothers were denied access to antiretroviral therapies. This has costed more than 300,000 lives [51].

These examples of disinformation share some thematic commonalities: assertions and preservations of power, authoritarianism, fearmongering, scapegoating to deflect blame, and the creation or reinforcement of states of collective shock which can be used to facilitate the implementation of political and economic agendas that are more difficult to achieve during periods of stability [52]. 

Misinformation, stigma, vaccine hesitancy and non-adherence to government recommendations seem to be in a vicious circle in which they influence, and are influenced by, each other.

During infectious disease outbreaks, individuals are recommended to comply with specific guidelines to prevent infection and reduce further disease transmission, such as self-quarantining, seeking medical treatment, and reporting contacts to public health officials. However, stigma can deter people from adopting these behaviours. This reinforces the link between misinformation and the tendency not to adhere to government recommendations [47], which was also confirmed by Lee et al., who found that COVID-19 misinformation exposure was associated with misinformation belief, and misinformation belief was associated with fewer preventive behaviours [53]. 

Other studies directly found that misinformation increases the rate of infected subjects in the population. Specifically, if measures are applied to contain the epidemic, the presence of misinformation has a negative impact on their effectiveness. When protection, distancing measures, and detection and isolation policies are not applied, misinformation increases the infection and death rates even in the presence of a full lockdown [54].

Moreover, it seems that greater worry and weaker endorsement of unfounded COVID-19 beliefs lead to more responsible behaviour [55].

#### 3.1.4. The Psycho-Socio-Demographic Profile of Misinformation

To outline the profile or to try to define a pattern of the personal characteristics associated both with the spread of and belief in misinformation is not an easy task. It is not surprising that the results of these investigations are conflicting. 

For instance, Lee et al., 2020, found that misinformation exposure was associated with a younger age, higher education levels, and lower income [53], while according to Agley et al., 2021, sociodemographic, political orientation and religious commitment are marginally, and are typically non-significantly associated with COVID-19 belief profile membership [56].

In contrast, an exploratory survey which was run to assess whether patterns of individual differences in political orientation, social dominance orientation and traditionalism may predict the willingness to share different kinds of misinformation regarding the COVID-19 pandemic online showed that liberals with a low disposition toward social dominance are specifically less willing to share conspiratorial misinformation than are conservatives with a high disposition toward social dominance, at least regarding a culturally salient scientific topic [57].

A clear relationship has also been found between right-leaning media consumption and pandemic-related public health beliefs: right-leaning outlets were more likely to make inaccurate claims about the origins and treatment of COVID-19, and people who self-reported the consumption of more right-leaning news were subsequently more likely to express misinformed views. In turn, misinformed individuals were more likely to think that public health experts over-estimated the severity of the pandemic [58].

Trust in science seems to be a strong, significant predictor of the adherence to irrational beliefs, with lower trust being substantively associated with misinformation to varying degrees [56]. Believing in the role of conspiracies, biological warfare, and 5G networks in the origin and spread of COVID-19 was associated with a higher level of anxiety, while the reliance on reliable sources to obtain information on the current pandemic was associated with lower levels of anxiety [59].

Furthermore, misinformation exposure is associated with psychological distress, including anxiety, depressive and post-traumatic stress disorder symptoms, as well as irrational beliefs [53]. 

Mejova and Kalimeri, 2020, checked whether the emotional content in misinformed advertisements on Facebook was different from the rest of the dataset, and they found no statistical significance, which may indicate that misinformation indeed “sounds” like the rest of the content [60].

### 3.2. The Dispositional and Situational Explanations of COVID-19 CTs

#### 3.2.1. COVID-19 CT and Socio-Demographic Elements: Education 

Several studies have shown that educational attainment is an especially important factor when it comes to the understanding and perception of conspiracy beliefs. Duplaga et al., 2020, analysed a sample of Polish internet-users, and found that younger age, lower educational levels and lower health literacy predicted the endorsement of a COVID-19 CT, but this association was reversed for internet-health literacy, which correlates with a higher probability to embrace a COVID-19 CT [61]. According to the data gathered from the Centre’s American News Pathway Project, which included 9654 US adults, almost half of the Americans with high school diplomas or lower education evaluated COVID-19 CTs to be probably/almost definitely true, but only 38% of those who had attended some college, 28% of those with a bachelor’s degree, and 15% of those with some postgraduate training endorsed these ideas [62]. The Italian data collected by Somma et al., 2020, confirmed the protective role of education that is positively associated with scientifically supported COVID-19-related causal beliefs, and negatively related to COVID-19 non-scientifically supported beliefs [63]. 

Surprisingly, the model of Stoica et al., 2020 found that a higher educational level is a positive predictor of adherence to COVID-19 CT. This apparently counterintuitive result may be explained by the particular social context of the study, which was conducted in Romania, where the trust in the government decreases as the level of education increases; therefore, it was hypothesized that the better-educated Romanians’ distrust in the government might explain those findings [64]. 

CT beliefs were found to be higher in those with only high school education [65]; however, there are suggestions that the effect of the level of education is small and not always protective, especially for those who have limited scientific knowledge that might lead to an illusion of competence [66]. 

#### 3.2.2. COVID-19 CT and Socio-Demographic Elements: Gender and Age 

According to previous studies concerning CT, gender does not seem to be a significant factor in influencing the adherence to COVID-19 CTs [64]. However, according to Schaffer, 2020, in the USA, women are slightly more likely than men (29% vs. 21%) to see at least some truth in the CT that powerful people planned the outbreak [62]. On the contrary, Cassese et al., 2020, showed that men are more likely than women to endorse COVID-19 CTs, and this finding was associated to men’s higher scores on learned helplessness scales, and their propensity for conspiracy thinking [67]. These results are consistent with the notion that gender does not influence COVID-19 CT endorsement per se, and that the gender gap may be explained by the fact that men scored higher for certain psychological predisposition factors in Cassese’s study. 

Considering age, younger people seem to be more likely to believe in COVID-19 CTs than older subjects [61,62,64,68]. 

Overall, it appears that age and sex can only explain minor differences in the probability of embracing COVID-19-related CTs. 

#### 3.2.3. COVID-19 CTs Turned Out to Be Closely Related to Other CTs

One of the most robust findings that emerged from the research on CTs is that believing in one CT predicts believing in others [69,70,71]. According to this point of view, several of the papers included in this review emphasized the importance of appreciating that the endorsement of a COVID-19 CT would be associated to broader CT beliefs [3,65,72]. For the current pandemic, a single unified narrative does not exist, but many COVID-19 CTs appear to be a coalescence of disparate domains of knowledge [73]. COVID-19 CTs have been embedded within millennial and apocalyptical thoughts, through different narratives that belong to distant and opposite cultural groups, such as the religious and secular, right-wing and radical-left [74,75]. These theories, even if they are separated by nuanced boundaries, share semiotics, shades and topics which all deal with a concern regarding the abuse of power [75]. 

Miller, 2020, explored the phenomenon of COVID-19 CT hyper-inclusivity, showing how a large majority believe in more than one CT, and that COVID-19 CTs are positively related to one another even if they are contradictory [74].

#### 3.2.4. Uncertainty, Stress and Anxiety 

There is extensive literature suggesting that belief in CTs is based on the epistemic need for certainty and control [76]. The risk to engage in CTs is higher during a societal crisis when people are seeking to make sense of a chaotic world [7,10,77]. It has been suggested that people often need somewhere to go with their anger and confusion; blaming and shaming may serve that purpose and make people feel less helpless [78]. From this point of view, it is quite intuitive to conceive the COVID-19 pandemic as a perfect storming event, causing vulnerability to conspiracy narratives that provide handy and captivating answers to the causes of an event of such catastrophic proportions; by providing the answers about the crisis and the actors behind it, they contribute to the restoration of a sense of predictability. 

Hopelessness, which is a result of the self-perceived inability to respond to external threats, stimulates a desire to create a simulation of safety and correlates significantly with COVID-19 CTs [79]. A high level of uncertainty (concerning oneself, one’s place in the world and one’s future) is a common factor for CTs, even if they are contradictory. In other terms, the COVID-19 belief system impacts more powerfully on those who experience a greater deal of situation-induced uncertainty [74]. 

Moreover, COVID-19-induced uncertainty bolsters the effect of personological predispositions and environmental risk factors in the facilitation of belief in a wide array of COVID-19 CTs [80]. Kim et al., 2020, analysed data from a survey of Korean citizens (*N* = 1525) and found that perceived-risk and anxiety have the most decisive impact on beliefs in CTs, together with the quality of information, health status, and support for President Moon Jae-Ins government [81]. 

Georgious et al., 2020, performed a study during the early period of the quarantine that did not confirm the association between people’s level of stress and conspiratorial beliefs. However, it is important to notice that, at that time, the sense of fear was well balanced with a collective motivation that somehow restored the sense of control and made people more receptive to health-related advices, so we can suppose that, during the specific time analysed by these authors, fear and uncertainty were not the primary inducing factors of CTs [65]. 

#### 3.2.5. Epistemic Untrust 

Within the social network world, COVID-19 is frequently presented in association with authority-questioning beliefs [29]. A widespread skepticism toward the official account of truth seems to be a shared element of distinct millennial groups and conspiracist groups with various political orientations, such as evangelical millennialism, the anti-vaccine movement, and New Age, left-wing and right-wing conspiracists [75]. Low levels of epistemic trust and a psychological predisposition to reject expert information and accounts of major events (denialism) predict the level of conspiratorial belief regarding the pandemic [82,83]. The same causal pattern is detectable also in the study by Oleksy et al., 2020, which analysed the relationship between a sense of control (individual or collective) and COVID-19 CT on a sample of 2726 Polish citizens, distinguishing two types of COVID-19 CTs: general CTs and government-related CTs. The study showed that a lack of an individual sense of control was associated with both types of CTs, while a feeling of collective control was positively related to general CTs but negatively to government-related CTs [84]. 

Mistrust toward the government also appears to be involved in the adherence to CTs in Korea and Romania [64,81]. 

A core element of distrust is in line with the previously expressed tendency of the conspiracy narratives to converge; this could be understood as a reflection of the underlying worldview in which the details of individual CTs are less important than a generalized rejection of official explanations.

#### 3.2.6. Political Affiliation 

Three studies conducted in the US found that voting Republican and embracing conservative views positively predicted beliefs in COVID-19 CTs [80,83,85]. The same relationship has been found in Turkey for religious and right-wing affiliation [72]. According to Shaeffer, 34% of Republicans and independents who lean on the Grand Old Party thought that the COVID-19 outbreak had been either probably or certainly planned by someone powerful [62]; political polarization emerges also from the study by Pennycook et al., 2020, in the USA and Canada [86]. In contrast, Stoica et al., 2020, found that far-right political views predict decreased beliefs in COVID-19 CTs, and that holding far-left political views, even if without reaching significance, has a positive influence on endorsing such CTs [64]. This observation about COVID-19 CTs is in line with previous ones: although COVID-19 conspiracy beliefs do not belong exclusively to the ideological right or left, if we consider an international overview, the asymmetry is marked by ideological processes that are nation-specific and more common in the extreme-wings [87]. 

The patterns of media use also show the clear influence of religious and political leaders on the acceptance of COVID-19-related unbased beliefs. Right-leaning outlets were more likely to make inaccurate claims about the origins and treatment of COVID-19, and people who self-reported a preference for more right-leaning news were subsequently more likely to think that public health experts overestimated the severity of the pandemic [58,88]. 

On Facebook, the trajectory of COVID-19 rumors seems to develop from small pre-existing conspiracist groups, and then is taken up into more diverse communities and receives substantial amplification by celebrities, evangelists, sports stars and media outlets [3]. The popular theories of #Filmyourhospital posit that the pandemic is a hoax, and suggest that hospitals are empty. Not only the majority of the users of the #filmyourhospital network but also the most influential ones, who fueled and expanded this conspiracy in its early days, were prominent conservative politicians and far-right political activists [47,89]. Pastor Chris Oyakhilome, the acclaimed religious leader, founder of Believers LoveWorld (a network of Evangelical churches with approximately 5,000,000 followers all over the world), claimed that the COVID-19 lockdown was a ploy to control the population, so the government could covertly deploy 5G, adding his voice to the conspirative group in Nigeria. Analysing his statement during the public debate, Ndinojuo et al., 2020, observed that it fell within the sphere of CTs, not necessarily because he was wrong, but because he failed to provide any empirical foundation to his claim [90]. The reluctance that the Brazilian government has had to impose the lockdown is well known; in Brazil, far-right political parties are closely linked to Evangelical churches, within a mutually dependent relationship, and Bolsonarism also used the religion as a political force during the pandemic [91]. The approach of Bolsonaro and his allied evangelical representation is an example of the minimization of the dangers of the pandemic that, even if it does not technically comply with the definition of a CT, shares its underlying paradigms and consequences in terms of health risk for the population. The reference to other infections, such as tuberculosis and flu, in comparison to COVID-19; the use of the number of deaths in the middle of the pandemic as a strategy to minimize it; the constant invalidation of the press; and the focus on buses and transport cleaning are known rhetorical techniques used to legitimize the fraudulent information that the disease is not so serious.

The press was accused of spreading panic, and a refined minimization attitude was applied with an argumentative strategy that resorted to rhetoric, allusion and unspecific religious themes such as “panic is devastating to the immune system, a poison”, “the church is an emotional hospital”, and “the blood of Christ is the greatest immunizing power that exists, we are covered by the blood of Jesus” [91]. This most likely created an overlap between the political and religious faith of the audience and their critical and rational capacity to interpret the events. 

#### 3.2.7. Personality/Psychological Traits

The most relevant theoretical perspective on the ideological character of conspiracy beliefs claims that beliefs in CTs comprise (or arise from) a conspiracy mindset, which is defined as the general tendency to view events as the product of a conspiracy, and it is the result of individual differences in thinking styles, cognitive ability, and motivation of critical thinking [7,8,87]. A conspiratorial thinking style results from a dispositional factor that is positively associated with COVID-19 conspiracy beliefs [72,80,83]. As expected, accurate beliefs about COVID-19 were broadly associated with cognitive reflection, and actively open-minded and analytical thinking [55,64,72,81,86]. Belief in COVID-19 CTs had a positive correlation with faith in intuition, and with lack of sensibility to a contradiction, both typical characteristics of irrational thinking [72,79]. 

Another psychological trait that is associated with CTs is the avoidance of uncertainty: those who are less tolerant of uncertain situations and consider themselves unable to respond to external threats (feeling hopelessness) were more likely to believe in CTs on COVID-19 [72,79,92]. Moreover, the research by Miller and colleagues also found that the psychological predisposition to resilience is a buffer against COVID-19 CTs [80]. This finding about resilience fits perfectly within the framework of the conspiracy mindset draw up by the cognitive and personological-style analysis joined with the uncertainty response patterns. 

COVID-19 CT has also been associated with moral character traits such as honesty/humility (with a negative association); extraversion [79]; machiavellianism and primary psychopathy (with a positive correlation), and external blame attitude [79,81,93].

Collective narcissism (an exaggerated sense of superiority of oneself, which extends to exaggerated sense of in-group superiority) is another psychological phenomenon that researchers have correlated with proneness to accept COVID-19-related CTs. A global health crisis may be perceived as a threat to one’s in-group national identity. Individuals with a heightened sense of in-group superiority may therefore perceive such pandemics as a threat to their national image, and may use conspiracies to manage a heightened hypersensitivity to these in-group threats [64,93,94].

### 3.3. The Behavioural Effects of COVID-19 CT 

#### 3.3.1. Sense of Control vs. Violent Behaviours

The study of Imhoff et al., 2020, showed that COVID-19 CTs gave people a certain sense of control; furthermore, Oleksy et al., 2020, demonstrated that the sense of a lack of individual control was associated to general and government-related COVID-19 CTs [84,95]. Another study indicated that in order to improve a sense of control, people could adopt magical thinking: one of the main expressions of this is pseudoscience [96]. Like COVID-19 CTs, pseudoscience made people able to fight the uncertainty originating from the actual pandemic. 

On the contrary, COVID-19 CTs could make people more inclined towards dangerous behaviours, making them more violent and hostile. The increase of violence related to COVID-19 CTs, in particular towards the authorities, was reported by other authors [63,83,97]. Violence could originate from feelings of hostility towards other people, and hostility itself could be related to the anger originating from the perception of an event—the pandemic—as an “inflicted injury” [98]. Maftei et al., 2020, found that believing in COVID-19 CTs concerning 5G was related to an increase in violent behaviours driven by a feeling of anger, especially in people more prone to paranoid thinking [97]. Oleksy et al., 2020, found that COVID-19 CTs were linked to xenophobic policy and negative attitudes towards people that came from countries associated with the spread of Coronavirus or other ethnic groups [84]. 

#### 3.3.2. Compliance with the Governments’ Rules

Governments, authorities and organizations like the WHO provided guidelines and behavioural instructions in order to reduce the spread of the virus and contain the pandemic, such as social distancing, auto-isolation, handwashing, respiratory hygiene, disinfecting surfaces, and wearing masks and other personal protective equipment (PPE) [17,63]. Unfortunately, compliance and adherence with them were (and still are) not always good [99]. One study showed that, in England, COVID-19 CTs were related to a lack of socio-political control and a perception of lower social status; that is to say that people who felt marginalized from society were more likely to mistrust authorities’ health guidelines [100]. Similar outcomes were presented by another study which highlighted that mistrust in authorities led to reliance on other health practices, such as alternative medicine or pseudoscientific practices (consuming garlic, drinking ginger tea, rinsing the nose with saline, using essential oils, following special diets, etc.), precisely because authorities did not support them [13]. Finally, people who rejected health preventive behaviours may have denied authorities’ good intentions rather than believing in the existence of undefined COVID-19 CTs related to the disease [84]. Interestingly, one study demonstrated that people with a conspiracy-prone mentality adopted preventive behaviours when they were not promoted and defended by an official authority [12].

#### 3.3.3. Intention to Get Vaccinated

CTs about medical topics determined behaviours that threatened health, such as the low adherence to screening programs, medical check-ups and vaccination campaigns. Pummerer and Sassenberg, 2020, highlighted that believing in COVID-19 CTs was linked to lower social distancing [101], while other authors reported that the most important consequences of conspiracy thinking were prejudice, social conflict, an increase in crime, and lower involvement in global issues and preventive health behaviours [63,72]. Several studies reported that people who adhered to COVID-19 CTs expressed their intentions not to get vaccinated for Sars-Cov-2 [85,102]; in particular, the study of Bertin et al., 2020, highlighted that CTs did not refer to potential dangers related to vaccines, but to a distrust in vaccines themselves [103]. 

#### 3.3.4. Socio-Demographic Elements

The fact of following or not following health guidelines and government’s behaviour rules, like adhering or not to COVID-19 CTs, also seemed to be related to socio-demographic factors. One study revealed that adopting health-protection behaviours was positively correlated to female gender and age [68]. Similar outcomes came from other studies [104,105]; women followed health protection behaviours better than men, while men revealed a higher frequency of non-evidence-based preventative behaviours [63]. 

Allington et al., 2020, also found out that younger and older people found it more difficult to adhere to health behaviours [68]; for older people, this was probably due to the fact that it was more difficult for them to give up routine habits and their already reduced social relationships, and also because they used digital devices less than other people. On the contrary, other studies revealed that the adherence to adequate preventive behaviours was positively correlated to age, probably due to increased age-related mortality [63,106]. Regarding occupational status, students showed the lowest adherence to health protection behaviours, while healthcare workers showed the highest. Maftei et al., 2020, reported that elderly people were more compliant to behavioural preventive rules than younger people, probably because of the fear of health consequences [97]. Romer et al., 2020, showed that younger people were more inclined to follow COVID-19 CTs, and therefore not to take health protection behaviours [85]. 

#### 3.3.5. Coping Strategies

Especially in the first months of the pandemic, many governments implemented lockdown measures in order to reduce the possible spread of the infection: the timing of the implementation and, consequently, the following results were various [92]. Lockdown measures created feelings of stress, anxiety, depression, and “feeling trapped”, especially in the first weeks of implementation; being able to control a situation would have allowed people to better cope with it. One study highlighted that emotion-focused strategies were more effective in coping with pandemics than those related to problem-solving strategies [107]. Another study showed that Germans mostly applied problem-focused strategies [108]; in particular, emotion-focused strategies were used more in the female gender. During the lockdown period, compulsive and hoarding behaviours frequently occurred: people stored masks, hand disinfectants, disinfection products, and food, in some cases until stocks were exhausted [74,96]. Miller, 2020, also demonstrated that believing in COVID-19 CTs about Coronavirus origins was positively correlated to hoarding behaviours, and negatively to preventive behaviours like hand hygiene or social distancing [73]. One study showed that food hoarding was influenced by the quarantine period, and by the fact that people did not want to go out shopping every day [108].

#### 3.3.6. Political Ideology

Different studies highlighted that political views could also influence people’s behaviours; in particular, one study highlighted that American people who supported republican ideology were more inclined to doubt scientific concerns about Coronavirus or to minimize risks connected to the pandemic [62]. Similar results emerged from another study, which showed that authoritarianism was associated with a general mistrust in the Coronavirus, blaming China for the Coronavirus pandemic, and to a lower likelihood of wearing masks [109]. Political conservatism was linked to COVID-19 CTs adherence, and so it may have had a negative impact on preventive behaviours by feeding distrust in science [74,99,110,111]. Conservatives seemed to believe that people were able to control the spread of the virus but, at the same time, they felt little responsibility to themselves and the others [112]. 

#### 3.3.7. Risk Perception

Risk perception is considered to be one of the most important factors in predicting people’s preventive behaviour during the Coronavirus pandemic [106]: one study showed that during the SARS epidemic people with a high-risk perception of the outbreak adopted preventive measures more than other people [113]. During the COVID-19 pandemic, risk perception was positively related with engagement in preventive behaviours [104]. On the contrary, a Chinese study did not find any correlations between risk perception and preventive behaviours [114]. 

Risk perception could be considered as a psychological attitude consisting in three dimensions—probability, severity and susceptibility—which could affect human preventive behaviour [105]. Risk perception is also defined as a cognitive answer towards an event or phenomenon, in this case the pandemic, which causes emotional symptoms like anxiety and depression [115]. Curiously, vulnerable groups, such as elders and people with many comorbidities, presented a low-risk perception [116], while people who lived with vulnerable subjects, like children and elders, showed a high-risk perception [106]. It was reported that anxious and fearful people who expressed excessive worries about the pandemic showed higher compliance with preventive behaviours than people who adhered to conspiracy thinking, which could affect health risk perception [72,97]. Risk perception and compliance to lockdown measures were found to be higher in women, older people and non-CT-believers [97]. Orte et al., 2020, also highlighted that, during pandemics, people’s behaviour could have been negatively affected by the so-called “optimism bias”, also known as the “overconfidence effect”, according to which people perceived themselves as being at lower risk of infection and infecting than others [115,117]. Interestingly, a Dutch study revealed that people who had not already tested positive for Sars-Cov-2 perceived a higher risk of infection for their friends and families than for themselves [108]. High risk perception and trust in science were positively and significantly correlated to compliance with health preventive guidelines and adequate behaviours [92,99]. 

#### 3.3.8. Emotional Responses

Anxious people complied with health recommendations better than other people, probably because they felt a stronger sense of responsibility towards the spreading of the pandemic [55]. Fear related to Coronavirus misinformation or the risk of contagion was positively correlated to health preventive behaviours [82,104], but it could also cause hoarding behaviours [82]. In general, both fear and anxiety could help determine adaptive responses related to self-protection [92]. Moreover, Jovancevic et al., 2020, underlined the role of the optimism–pessimism dimension: it emerged that optimistic people were more able to put in place coping strategies, while unrealistic optimism was accompanied by the low probability to engage in preventive behaviours [82]. If on the one side wearing masks and social distancing needed a personal sacrifice, on the other they were connoted by an important pro-social meaning: indeed, adhering to them implied the emotional process of empathy towards vulnerable subjects. Pfattheicher et al., 2020, confirmed what previous studies already found: empathy promoted adherence and motivation to preventive behaviours, even though motivation was already high and beyond the perceived personal vulnerability [118,119].

#### 3.3.9. Social Media

Social media, television, media personalities and politicians played a considerable role in determining negative health preventive attitudes [68,111], although behavioural effects were controversial, probably due to the contents’ heterogeneity [63]. Indeed, in the early pandemic period it was difficult for people to adopt preventive behaviours, probably because there was a gap between healthy public messages and those circulated by social media [85]. Positive correlations between the fact of relying only on social media or non-scientific platforms and non-evidence-based behaviour were documented [63]. Wearing masks, social distancing and lockdown measures were fundamental elements in buffering the negative effects of misinformation: by not applying them, the virus was spread and deaths increased [54].

#### 3.3.10. Cultural Elements

People’s preventive behaviours are also affected by their cultural orientations, e.g., individualism and collectivism. It was highlighted that individualism, which is reflected in narcissistic traits, feelings of entitlement and feeling special, was associated with believing in COVID-19 CTs, and with a lower compliance with preventive behaviours than collectivism. Indeed, collectivism was related to the principle of sacrificing personal good for the benefits of all [120]. A Polish study highlighted that about 20% of the Polish population dedicated more time to religious practices during the Coronavirus pandemic than before, maybe because religion was felt to help people in facing life difficulties and accepting injustice, giving a sense of hope and protection [121,122,123]. The Polish studies also confirmed what a previous research had already suggested: religious people could easily violate national heath recommendations because they had little confidence in science. The increase in religious practices was positively correlated to the support of COVID-19 CTs [124].

## 4. Limitations

One key limitation of this study is that many of the papers reviewed were not experimental studies, but correlational analysis; therefore, the cause–effect direction is not clear, and might be multi-directional. A great deal of the studies used convenience samples, and often the recruitment methods, based on the administration of online questionnaires, indirectly excluded the part of the population that is less computer-literate or does not have access to computers. 

## 5. Discussion

This article examined CTs regarding COVID-19, with a particular focus on the link between them and the contextual events. We began by providing an overview of the information/disinformation flows and their relationship with the spread of these theories, sketching a summary of the dispositional and situational explanations of developing and embracing COVID-19 CTs by examining the behavioural consequences of the conspiracy thinking (see Table 1). 

The misinfodemic could be motivated in different ways: an epistemic desire for a causal explanation and subjective certainty, an existential desire for control and security, and the social desire to maintain a positive image of the self or group [125]. The COVID-19 infodemic has induced a confidence crisis and distrust in authorities, science communities, governments and institutions. It could lead to harmful health behaviours and worsen mental health, and could become as serious a threat to public and global mental health as another kind of virus [126]. 

The relationship between public trust and mental health is complex and bidirectional: the Coronavirus pandemic brings high levels of uncertainty, and the inability to cope with uncertainty is exponentially higher causing anxiety, fear and health-related worrying. This situation is worsened by the exposure to an overload of information, including misinformation and disinformation, from mainstream media and social media. Perceived threat information elevates fear, and repeated engagement with trauma-related stimuli ends in acute stress and emotional distress, and consequently will either increase the fear of the virus, or cause insensitivity towards its course [125].

The infodemic overabundance associated with multiple, dissonant and conflicting mental models makes it hard for many people to find trustworthy sources and reliable guidance in the COVID-19 crisis of confidence. Due to the infodemic, everyone has a private opinion and alternative truths, people commonly become confused, irrational, anxious, fearful, suspicious, xenophobic, even psychotic, and prone to extreme behaviours. 

The examination of the predictors of individual beliefs, such as personality and cognitive variables, is an attempt to understand the framework of fragility that boost individual beliefs, but to make a fuller sense of what is happening with COVID-19 CT it is necessary to explore not only psychological and socio-demographic elements, but also their underlying structure, analysing them in from a socio-political and historical perspective.

This review provides evidence of trans-national differences not to be overlooked, as political affiliation, and cultural and historical background invariably influence the antecedents, contents and consequences of irrational thinking. In particular, individuals who are in a larger and connected cultural context are expected to be less likely to endorse CTs concerning COVID-19, as the participation in a different context exposes people to different points of view and sources of information, which can decrease an individual’s tendency to believe in COVID-19 CTs. 

Health behaviour guidelines provided by the governments, such as the lockdown, were accompanied by multiple deprivations in the private, social and work spheres; the indirect effect of the pandemic on the economic situation has been dramatic for some groups of workers in particular, and this understandably influenced the capacity to accept reality. Taking this into account, it is easy to understand how social-environmental elements such as the economic situation could become a variable for embracing CTs [61]. 

COVID-19 CTs affected people’s behaviour in ways besides thinking. In many circumstances, COVID-19 CTs not only influenced the choice of unsafe health behaviours, but also caused threatening social side effects, such as the increase of violence and the spread of xenophobic policies. In addition to the personal factors, like age, gender and emotionally driven attitudes, cultural and socio-political contexts seemed to strengthen negative COVID-19 CTs’ related behaviours. 

Philosophy has offered interesting contributions to help understand the adoption of irrational beliefs or CTs, on the basis that, as irrational beliefs, they may provide a useful sense of control, restoring the epistemic capacity, even if at a very high cost and with potentially serious health consequences [5]. COVID-19 CTs play out along the lines of xenophobia and racism, and show the global pandemic not as biology, but as the manifestation of political affiliation, difference and disconnection [127]. A relevant philosophical contribution may derive from TOK, which can represent a useful tool to scrutinize knowledge and underline the importance of focusing on justified beliefs rather than emphasizing the infodemic [128,129]. 

Table 1 summarizes the main elements that can contribute to support COVID-19 CTs.

## 6. Conclusions

Coronavirus misinformation is the most important factor in delaying the adoption of the correct preventative behaviours to fight the global pandemic. As misinformation especially came from social media, policy makers, healthcare workers and experts should encourage social media campaigns that could promote a culture of fact-checking [59]. The practice of banning social media posts or suspending accounts spreading misinformation is a controversial one, but it is crucial to devise methods to identify credible influencers who promote appropriate information [31,36,111]. Other possible strategies include enhancing and promoting e-Health, and constantly analyzing patterns of exchange and sharing information on the web [130]. 

Promoting a sense of collectivism, being empathic and compassionate towards vulnerable and sick people, avoiding preconceptions about the link between the Coronavirus and specific ethnic groups, adopting “open-mind” thinking, and always verifying the source of information have been proposed as ways which could help to reduce misinformation and promote preventative behaviours and inter-communication [40,55,63,120].

In order to reduce negative behaviours related to fear and the distortion of risk perception, a list of effective coping strategies was produced, such as practicing mindfulness, doing physical activity, and planning long-term goals for minimizing stress effects; the usefulness of maintaining a daily routine, especially for people in quarantine, was also reported [40,107,115].

An important goal in facing pandemics is to improve resilience against uncertainty: this could be achieved through many methods, such as enhancing a mindful personal attitude (focusing attention over the present moment), social support on digital platforms, physical activity, and spiritual health [74]. Resilience could be strengthened and taught; in this process, every part of society is essential, but especially mental health care professionals, who should invest more in the research of psychoeducation as a possible way to promote trust [126]. Trust promotes interpersonal collaboration, especially in fragile scenarios, and is positively correlated to compliance with health recommendations [82].

Governments and authorities should promote clear and effective communication strategies to reduce social media misinformation and improve discussion, in order to better understand the pandemic [106,131]. They should also communicate using reliable social media and newspapers, and give person-centered answers to the problems which the Coronavirus pandemic poses [40]. One study demonstrated that deontological messages had a greater impact than utilitarian and virtue-based ones in delivering public health messages, and in determining preventative behaviours, regardless of the source [112]. 

In order to fight the COVID-19 pandemic, a precious point of view to adopt could be that of “think globally, act locally” [132]. The global leadership should approach the actual world health crisis through international cooperation, empathic communication and solidarity, because no country can do this alone. A crucial feature that Governments should own is that of the humanistic vision, characterized by trust, the promotion of public relationships, and creative communication [126]. 

Moreover, mental health promotion could change the impact of the COVID-19 pandemic by creating and enforcing a humanistic self and an empathic civilization [126,132]. Finally, health care professionals should know that spreading health information, participating in group supervisions, and staying up to date with new and trustworthy guidelines are among their responsibilities as clinicians [40]. 

## Figures and Tables

**Figure 1 ijerph-18-09839-f001:**
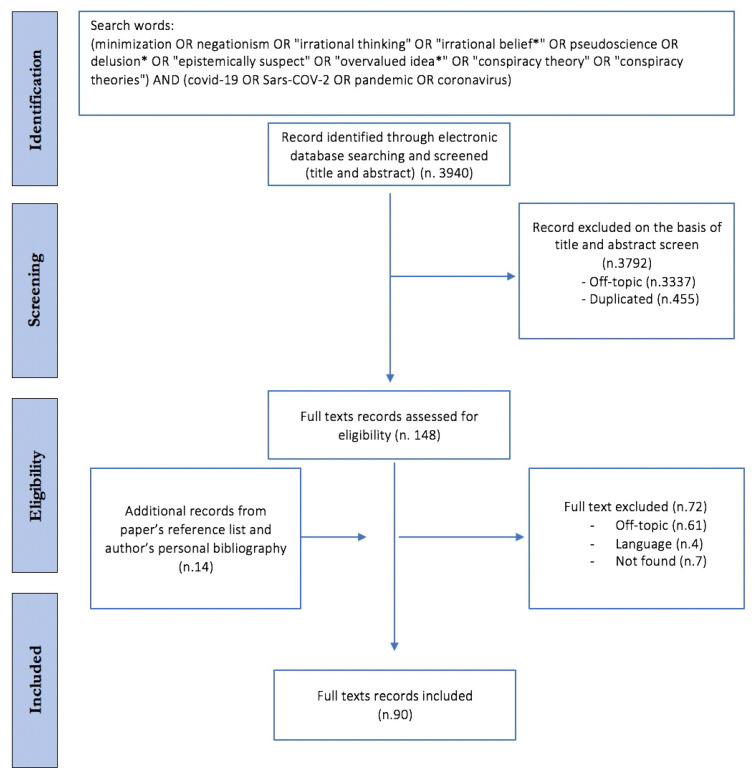
Prisma 2009 flow diagram: selection of included studies. Abbreviations: COVID-19 (coronavirus disease 2019); Sars-COV-2 (severe acute respiratory syndrome coronavirus 2).

**Table 1 ijerph-18-09839-t001:** Overview of the dispositional and situational explanations of COVID-19 CT endorsement.

Socio-Demographic Elements	Situational Elements		Personality/Psychological Traits
	Contextual	Antecedents	
Low educational level Limited scientific knowledge Younger age No gender difference	Experiencing situation-induced uncertainty High level of self-perceived risk and anxiety	Believing in other CT not related to COVID-19 Mistrusting toward the government Political or religious polarization Self-perceiving inability to respond to external threats	Low levels of epistemic trust Predisposition to reject expert information Conspiracy prone mindset Avoidance of uncertainty Low resilience predisposition Extraversion Machiavellianism and primary psychopathy External blame attitude Collective Narcissism Cognitive characteristics: - faith in intuition - resistance to contradiction

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
