# Peer review of "Irrational Beliefs about COVID-19: A Scoping Review"

_ijerph, 2021, doi:10.3390/ijerph18199839_

Round 1

Reviewer 1 Report

The paper is an excellent and comprehensive scoping review of the scientific literature about COVID-19 related misinformation and conspiracy theories. I would suggest to the authors to discuss their results in the frame of theory of knowledge (TOK). In our world of infodemics where sources of information are plentiful and uncontrollable, TOK can provide us with the tools  we need to scrutinize knowledge and identify the truth. In the introduction it would be useful to define rational thinking (beliefs) and irrational thinking (beliefs). For example, rational thinking can be defined as thinking that is consistent with known facts, and irrational thinking as thinking that is inconsistent with, or unsupported  by known facts, while critical thinking is the systemic evaluation or formulation of beliefs, or statements, by rational standards. Regrettably, irrational beliefs are commonly the primary reason for human misery and dysfunction so explaining their psychodynamics, individual and collective, could help in overcoming conspiracy theories and their detrimental health, social and political consequences.  

Reviewer 2 Report

The authors present an excellent overview of Covid-19 related conspiracy theories, their correlates and boundary factors. The topic is timely and of importance; the overview appears to be well conducted and balanced; and the writing is concise and informative. I have only few minor points listed below.  

Minor issues 

- Affiliation: "School of Specialization in sychiatry": "Psychiatry" 

- l. 75: "to explain people social behaviours": "people's"?

- l. 135: Where studies assigned to several groups when they dealt with more than one topic? Did the authors use other review papers as a starting point / source of references?

- l. 139: "The software used for the data collection was Excel": I am not sure this is a relevant information.

- l. 144: "A meta-analysis was not conducted": Maybe add "quantitative"?

- l. 436: "#filmyourhospital network": A short explanation could be added; same for "Believers LoveWorld2 (l. 439)

- Table 1: Start on new page

- l. 725: "that could promote a culture of act-checking": "fact-checking"?

- Appendix A (references from the searches): To save space, this list could be removed / integrated into the general list of references (contributions considered in this review marked with an asterisk).
